# Novel Molecular Methods in Soft Tissue Sarcomas: From Diagnostics to Theragnostics

**DOI:** 10.3390/cancers17071215

**Published:** 2025-04-03

**Authors:** Nicholas Frazzette, George Jour

**Affiliations:** Department of Pathology, NYU Grossman School of Medicine, New York, NY 10016, USA; nicholas.frazzette@nyulangone.org

**Keywords:** sarcoma, molecular pathology, genomics, epigenomics, transcriptomics

## Abstract

Soft tissue sarcomas (STSs) are rare cancers that arise from connective tissues like muscles, fat, or blood vessels. They can be challenging to diagnose because of their wide variety and overlapping features under the microscope. Our review explores the use of advanced genetic and molecular tools to improve the diagnosis, classification, and treatment of these tumors. By analyzing DNA and RNA from tumor cells, novel methods aim to uncover unique genetic changes that can guide personalized treatments, predict patient outcomes, and even discover new drug targets. This review highlights how integrating cutting-edge technologies into sarcoma care can lead to more accurate diagnoses and better treatment strategies, ultimately benefiting both patients and the broader medical community.

## 1. Introduction to Genomic Classification of Soft Tissue Sarcomas (STSs)

Soft tissue sarcomas (STSs) are a diverse group of malignant tumors derived from mesenchymal tissues. Historically, STSs were classified histologically based on the normal mesenchymal tissue they most resembled. This classification scheme led to more than 70 subtypes of STSs, each with distinct pathological and clinical features. However, accurate histologic diagnosis can be challenging as there is a significant degree of pleomorphism and morphologic overlap. In one multicenter study, the authors found that nearly one quarter of sarcoma diagnoses initially based on histomorphology and immunoreactivity were revised following molecular profiling [1]. Additionally, while histologic diagnoses have been correlated with prognoses, inaccurate or descriptive diagnoses can preclude medical decision-making, underscoring the need for advanced diagnostic and theragnostic testing.

Genomically, STSs can be broadly divided into two categories based on their karyotypic complexity: simple karyotype and complex karyotype. Simple karyotype sarcomas often harbor specific genetic alterations, such as translocations or activating mutations, which are key diagnostic markers and therapeutic targets. In contrast, complex karyotype sarcomas lack a specific genetic signature and exhibit numerous chromosomal abnormalities, reflecting a high level of genetic instability. While this complex karyotype may predispose certain carcinomas to highly aggressive behavior, they can also provide diagnostic utility, and molecular profiling may still reveal clinically actionable genomic variants that can slow disease progression.

In this review, we first present an update on the various molecular techniques used in STS profiling with an emphasis on their diagnostic and theragnostic approaches. Additionally, we discuss the various genomic platforms using tissue-based and liquid-based analytes. Finally, we solidify the discussion of these molecular techniques by reviewing selected known molecular drivers of various STSs, their detection by different molecular techniques, and their diagnostic and therapeutic importance.

## 2. Current Molecular Techniques in Sarcoma Diagnosis and Theragnosis

Molecular profiling is pivotal for the accurate diagnosis, classification, and therapeutic stratification of sarcomas. Histologic diagnosis is challenging owing to a high degree of pleomorphism and morphologic overlap among STSs. The wide variety of molecular profiling techniques that have been developed in the field of STS pathology underscores the clinical need: targeted profiling of histologically simple cases can yield therapeutic information while wider profiling can elucidate diagnoses in histologically challenging cases, or even lead to the development of new molecularly defined tumor subtypes [2,3]. The techniques used can be DNA- or RNA-based and are essential in uncovering genetic and epigenetic changes that drive these malignancies. Recent editions of the World Health Organization (WHO) Blue Book series recognize this fact, as STSs are increasingly becoming molecularly defined. Applying either a wide or targeted lens to DNA sequencing arrives at whole-genome sequencing (WGS), where the entirety of the genome including exons and introns are probed, whole-exome sequencing (WES), where only the exonic regions of the genome are probed, or targeted panels where only a number of specific genes are probed. Similarly, applying the same lenses to RNA sequencing arrives at whole-transcriptome sequencing (WTS), where all the post-transcription processed exonic sequences are probes, or targeted panels, where only a number of specific transcription products are probed. However, each technique has its strengths and pitfalls, as summarized in Table 1, requiring a thorough understanding to be able to appropriately deploy each technique to achieve the most utilizable results.

### 2.1. DNA-Based Techniques

Amplicon-based sequencing assays involve the targeted amplification of specific genomic regions of interest using primers and subsequent high-throughput sequencing, making them ideal for detecting known mutations or variants in a focused manner, such as in microbial genomics or cancer panels. In contrast, hybridization capture-based sequencing assays employ labeled probes that hybridize to target sequences in a complex nucleic acid mixture, followed by capture and subsequent sequencing; this method allows for broader and more flexible targeting across larger genomic regions and is widely used in whole-exome sequencing and larger, more comprehensive gene panels. Together, amplicon-based sequencing and hybridization capture-based sequencing constitute the main technologies used in next-generation sequencing (NGS) and provide complementary capabilities in terms of sensitivity, target specificity, throughput, and scalability, allowing for a range of applications from highly targeted studies to comprehensive genomic profiling.

The IonExpress and Illumina platforms are both commercial next-generation sequencing (NGS) technologies that employ different sequencing methods but can be used with amplicon-based and hybridization capture-based sequencing assays, respectively. Illumina technology, which relies on sequencing by synthesis (SBS), utilizes reversible terminator bases to incorporate labeled nucleotides one at a time, allowing for highly accurate and massively parallel sequencing. Illumina platforms are compatible with both amplicon-based and hybridization capture-based assays, enabling high-throughput sequencing of specific genomic regions or broader genomic targets, respectively. IonExpress, developed by Thermo Fisher Scientific, is part of the Ion Torrent suite of sequencing technologies and utilizes semiconductor sequencing technology, where nucleotide incorporation is detected by measuring hydrogen ions released during DNA synthesis, offering a faster and cost-effective option with moderate read lengths. IonExpress is particularly suited for amplicon-based sequencing due to its rapid run times and lower costs, but it can also perform hybridization capture-based sequencing for more extensive genomic analyses. Thus, while both Illumina and IonExpress are compatible with these sequencing methods, they differ in terms of their underlying sequencing chemistry, cost, accuracy, read length, and suitability for specific applications, providing researchers with options tailored to their needs in genomics research.

#### 2.1.1. Circulating Tumor DNA (ctDNA) and Cell-Free DNA (cfDNA) Diagnostics

The use of circulating tumor DNA (ctDNA) and cell-free DNA (cfDNA) in molecular diagnostics is emerging as a transformative approach for soft tissue sarcomas, offering insights into tumor biology and therapeutic management. cfDNA is cell-less DNA and can be used to describe any DNA material recovered from any cell, benign or malignant. ctDNA consists of fragmented tumor-specific genetic material released into the bloodstream by cancer cells through processes like apoptosis, necrosis, and active secretion [13]. cfDNA can be used for noncancer uses, such as prenatal testing or organ transplant surveillance. However, for the purposes of this review, focused exclusively on malignancies, the two terms are used interchangeably. Because ctDNA and cfDNA can be collected with a simple blood draw, they provide a non-invasive alternative to traditional biopsy, enabling the detection of genetic alterations associated with sarcomas. Professional society guidelines, such as those from the National Comprehensive Cancer Network (NCCN) and European Society for Medical Oncology (ESMO), increasingly recognize these so-called “liquid biopsies” as adjunctive tools for clinical trials, companion diagnostics, and disease monitoring, although they stop short of endorsing them as primary diagnostic methods due to current limitations in standardization and sensitivity. In soft tissue sarcomas, ctDNA and cfDNA analyses are especially valuable given the heterogeneity and often challenging biopsy locations of these tumors [14]. Meta-analyses, such as those by Kelly et al. (2021), underscore the utility of ctDNA for detecting actionable mutations (e.g., KIT and PDGFRA mutations in gastrointestinal stromal tumors) and assessing response to tyrosine kinase inhibitors [15]. Case reports have further demonstrated that ctDNA can reveal mutations in sarcoma subtypes like Ewing sarcoma [16] or leiomyosarcoma [17], facilitating tailored therapies where tissue biopsy might otherwise be impractical or yield insufficient material.

Comparisons between liquid biopsy and traditional methods reveal complementary advantages. Tissue-based diagnostics remain the gold standard for initial histologic classification, immunohistochemistry, and specific genetic rearrangements. However, ctDNA offers superior potential for longitudinal monitoring and detecting tumor heterogeneity, as it reflects genetic alterations across primary and metastatic sites [18]. For example, a recent case series demonstrated how ctDNA tracking enabled real-time mutation profiling and treatment adjustments in synovial sarcoma patients undergoing treatment [19]. Nonetheless, challenges persist. ctDNA assays often struggle with low tumor fraction in plasma, especially in less aggressive sarcomas, necessitating highly sensitive technologies such as droplet digital PCR or next-generation sequencing [20]. Moreover, cfDNA can arise from non-tumor cells, complicating interpretation without adequate controls [14]. In some cases, such as with Signatera liquid biopsy for minimal residual disease monitoring, the initial liquid biopsy is paired with a tissue sample to increase specificity; subsequent monitoring can be conducted with just liquid biopsies. These challenges are overcome as protocol standardization and probe specificity improves, though, and currently there are several commercial liquid biopsies available or pending approval in the United States (Table 2).

Despite hurdles, the versatility of ctDNA profiling is undeniable, with ongoing trials exploring its integration into multi-omic platforms alongside proteomics and metabolomics. Clinically, ctDNA is particularly promising for high-grade sarcomas, where rapid detection of resistance mutations could guide therapy modification. With advancements in bioinformatics and assay validation, the role of ctDNA in soft tissue sarcoma diagnostics and management is poised to expand, bridging the gap between innovation and routine clinical use. Currently, thirteen clinical trials in soft tissue and bone sarcoma are utilizing cfDNA or ctDNA for pathogenesis characterization, diagnostic inclusion criteria, and/or treatment response and disease progression monitoring, as summarized in Table 3.

#### 2.1.2. Whole-Genome Sequencing (WGS)

WGS represents the most comprehensive sequencing technique, sequencing the entire genome including exons and introns. This comprehensive approach allows for the detection of both coding and noncoding variants. While the comprehensive nature of WGS covers the detection of known variants, it generates tremendous amounts of data and can commonly detect a high number of variants of unknown significance. Therefore, it requires significant laboratory professional and pathologist curation and may not represent the most efficient diagnostic tool. However, WGS can potentially further optimize the clinical care of sarcoma patients, because germline DNA also is sequenced as part of the diagnostic workup. The use of paired germline and tumor DNA sequencing for somatic variant calling in WGS enables the detection of previously unrecognized pathogenic germline variants.

WGS has significant clinical implications in sarcoma management. In a prospective cohort study at a tertiary referral cancer center, WGS provided direct clinical utility in 24% of sarcoma cases, resolving diagnostically challenging tumors, identifying new treatment options, and detecting previously unrecognized germline variants. Importantly, WGS prompted a revision of diagnosis in 14% of cases, leading to changes in the treatment plan in several instances. Additionally, pathogenic germline variants were identified in approximately 8% of patients [4]. In another prospective study at a specialized sarcoma treatment center over a two-year period, incorporation of WGS as a diagnostic standard led to diagnostic refinement in 37% of cases and identification of therapeutically targetable variants in 33% of cases [5]. A further prospective study using the WGS of 200 soft tissue and bone tumors suspicious for malignancy led to the reclassification of 7% of diagnoses, either downgrading from low-grade to benign or reclassifying as metastatic malignant melanoma. Furthermore, treatment-relevant variants were detected in 15% of cases and germline variants associated with hereditary cancer syndromes in 11% of cases [21]. A retrospective study of all sarcoma cases at a specialized sarcoma center between May 2021 and September 2022 revealed two cases with germline mutations, with patients subsequently referred to clinical genetic services, and 47% of cases harboring variants with Association for Molecular Pathology tier 1A (biomarkers that predict response or resistance to approved therapies) significance [22]. These studies demonstrate the value of WGS in sarcoma management. Retrospectively, such studies can help revise diagnoses, unlock new therapeutic avenues for management, and uncover germline mutations for additional testing. Prospectively, these studies have demonstrated the importance of WGS in diagnostically challenging cases and ensuring patients receive the most effective targeted therapies.

#### 2.1.3. Whole-Exome Sequencing (WES)

WES focuses only on the ~1.5–2% of the human genome that represents coding regions, allowing for the identification of clinically relevant variants at much greater sequencing depth, although it provides less comprehensive coverage compared to WGS. Though coverage is less comprehensive, most sarcomas with known genetic characterization are driven by loss of tumor suppressor function or gain of oncogene function, both of which would be detected by WES. In a similar way to WGS, the use of matched tumor and normal tissue specimens in WES can be used to detect germline variants as well as somatic variants in STS. In addition, the more focused sequencing trims the overall amount of data to reduce cost and need for professional review, allowing for a more routinely available and clinically viable platform.

#### 2.1.4. Targeted Exome Sequencing

This approach is used to assess specific genes of interest, allowing for a focused analysis of variants known to be associated with sarcomas. Targeted sequencing often comes in the form of a panel of genes known to be the more common oncogenic drivers of STS. In one study, the use of a targeted panel led to the diagnostic refinement or reassignment of over 10% of study cases and the detection of potentially therapeutically targetable variants in nearly one third of study patients [3]. Targeted sequencing allows for much higher throughput and more cost-effective analysis at the tradeoff of the least comprehensive sequencing method. Similar to the tradeoff between WGS and WES, a targeted exome sequencing approach also allows for even greater depth of coverage so that, while the genes covered are the least comprehensive, the accuracy of variant calling in those covered genes is high. For example, a comparative study of WGS and targeted sequencing of 60 cases of adenocarcinoma identified 120 variants by targeted sequencing as compared to 114 variants by WGS. However, the study authors noted that WGS also provided richer genomic detail, including copy number variation, tumor mutational burden, and driver mutations detected in regions not covered by the targeted panel [23]. Despite these results, a targeted panel may often be best suited for clinical deployment, for example, focusing evaluation of a tumor for known oncogenic drivers which have therapeutic targets and limiting the detection of variants of unknown significance, or in situations where high-throughput demand necessitates a cost-effective approach.

### 2.2. RNA-Based Techniques

#### 2.2.1. RNA Sequencing (RNA-Seq)

RNA-Seq, also called transcriptomic sequencing for its emphasis on the transcribed genetic product, enables the detection of gene fusions, mutations, and expression profiles that are important in sarcoma subtyping. RNA-Seq, as compared to DNA-Seq, is particularly well suited for the detection of gene fusion products, and modern methods are capable of detecting fusions where only one partner, the oncogenic gene, is known. Advanced RNA extraction techniques, which use protocols like Trizol-based RNA extraction or the Qiagen RNeasy FFPE kit, have enabled the use of formalin-fixed paraffin-embedded (FFPE) samples and widened the use cases for this technology. RNA extraction of FFPE samples enables much more comprehensive testing, including the option of sending out samples to specialized labs or recovering RNA from archived tissues for long-term diagnostic characterization or research purposes. RNA-Seq can be performed broadly (so-called “whole transcriptome” sequencing) or with a targeted panel, with similar pros and cons to broad and targeted DNA techniques, as previously discussed. Indeed, targeted RNA panels have significant clinical implications. In one study, targeted RNA profiling of uterine sarcomas revealed a molecularly homogeneous subtype of high-grade endometrial stroma sarcoma with distinct sonic hedgehog pathway and *NTRK3* activation, with implications for immunohistochemistry diagnosis and endocrine therapy [24]. In another study, targeted RNA profiling of 153 carcinomas and sarcomas identified an additional 30 fusions not detected by other means, seven of which led to revision of diagnosis and 19 of which were novel and not previously described in the literature [25]. Use of targeted panels in these research and discovery contexts has led to significant effort in developing novel assays to detect common gene fusions for diagnostic assistance when histomorphology and immunohistochemistry are inconclusive, as often is the case in sarcoma pathology [26]. In parallel with these developments, several commercial RNA panels have been developed and approved for diagnosis, treatment planning, and disease monitoring, as summarized in Table 4.

In addition to detecting fusion gene products, RNA-Seq provides a window into the functional status of sequenced specimens by focusing only on the actively transcribed, and therefore expressed, gene product. Correlation of this cellular function with histologic diagnosis is a promising field of research with clear clinical utility in histologically indistinct lesions. For example, in a study of the heterogeneous class of small round blue cell sarcomas, a random forest machine learning algorithm trained on gene expression from a targeted RNA panel proved capable of predicting *CIC* gene rearrangements more accurately than the currently accepted immunohistochemical surrogate [27]. Such studies also demonstrate the potential importance of a wide transcriptomic fingerprint, rather than single variant events, in molecularly classified STS.

#### 2.2.2. Spatial Transcriptomics in STS

Spatial transcriptomics is a cutting-edge technology that analyzes RNA sequencing, often at single cell resolution, in combination with the location of the transcribed product in tissues [28]. Compared to bulk RNA sequencing of homogenized tissue, spatial transcriptomics can provide insight about the tumor microenvironment and cell heterogeneity at a much finer resolution [29,30]. Such analysis can provide insights into cell–cell interactions, effects of the transcriptome on tissue architecture, and mechanisms of progression such as invasion or metastasis. The earliest forms of spatial transcriptomic profiling involved laser microdissection of single cells from tissue, followed by single cell sequencing and then tissue reconstruction. While relatively straightforward, this technique is limited in throughput at the single cell level [31]. One simple innovation to laser microdissection was to carve off geographic regions rather than single cells to increase throughput at a trivial loss of spatial resolution [32]. Further advancements relied on in situ hybridization and in situ sequencing of target RNA within cells by complementary oligonucleotides that could be tagged and detected digitally [33,34]. While initially limited in both spatial resolution and number of RNA sequencing targets, advancements in fluorescence technology, digital imaging technology, and hybridization technology have enabled in situ-based spatial transcriptomics to achieve super-resolution imaging of thousands of genes [35,36,37,38,39,40,41]. Still, successive rounds of hybridization and imaging lead to increasingly complex and time-intensive analysis pipelines. The most recent advancements in spatial transcriptomics involve indexing or tagging tissue regions based on morphologic appearance, homogenizing the tissue and sequencing in a method like bulk RNA-Seq. However, by using nucleotide index tags that are sequenced alongside the extracted RNA, the transcriptomic product can be correlated back to the morphologic source [42,43]. As with other spatial methods, technical advancements to the conceptual method of spatial indexing have improved throughput and resolution [44,45]. These techniques are summarized in Table 5.

The application of these spatial transcriptomic techniques to sarcoma profiling is a novel and promising field of research for discovering diagnostic markers and therapeutic targets in a variety of STS [46]. For example, spatial transcriptomic profiling of the immune microenvironment in undifferentiated pleomorphic sarcomas led to one case of PD-1 blockade reducing tumor markers and eliminating metastases [47]. In malignant peripheral nerve tumors, spatial profiling identified a population of mesenchymal neural-crest-like cells, the quantity of which were found to correlate with a histologic grade and may have therapeutically targetable vulnerabilities [48]. Additionally, in gastrointestinal stromal tumors, the use of digital spatial profiling in a mouse model treated with imatinib showed high expression of T cell exhaustion markers in the immune microenvironment, potentially revealing checkpoint blockade as a therapeutic avenue [49]. Moreover, the differential presence of macrophages in the GIST microenvironment was found to correlate with high-risk tumor behavior [50].

### 2.3. Methylation Analysis

Epigenetic profiling, specifically methylation analysis, has recently emerged as a powerful diagnostic tool. Rather than sequencing genetic material directly, this method assesses the methylation status of CpG loci across the genome. Different tissues, both normal and pathologic, will display different methylation signatures that are consistent within a tissue or tumor class. Methylation data are particularly appealing because of their relative simplicity compared to genetic sequencing data. That is, a particular CpG locus can have only a binary state of being methylated or unmethylated, whereas a particular base pair locus can have one of four states. These binary data can be analyzed with relatively less computational resources than quaternary data. Such simplified input data can allow sarcoma subtypes to be distinguished with high accuracy by the use of a simple random forest machine learning algorithm based on a summation of thousands of binary methylation signals. A recent study validated this approach, showing that a classifier trained on 1077 methylation profiles accurately sorted a large, diverse cohort of sarcoma cases into 62 tumor methylation classes. The classifier was further validated on a cohort of 428 sarcomatous tumors and successfully classified over three quarters of this cohort [51]. Methylation analysis has potential for both research and future diagnostic applications.

## 3. Molecularly Driven Targeted Therapies in STS

The elucidation of specific genetic alterations in STS has enabled not only enhanced diagnostic accuracy, but also prognosis and therapeutic research. In particular, the development of targeted therapies, such as tyrosine kinase inhibitors (TKIs), has only become more feasible with greater and greater definition of the molecular aberrations that drive many STS pathogeneses. For example, tumor cells in tenosynovial giant cell tumor (TGCT) overexpress the growth factor colony-stimulating factor 1 (CSF1) to drive cancerous proliferation. This overexpression results from a translocation involving the CSF1 gene on chromosome 1p13 and the COL6A3 gene on chromosome 2q37. However, such overexpression can be neutralized by therapeutically targeting the CSF1 receptor with antagonist inhibitors such as pexidartinib, a small molecule TKI. In another case, approximately 90% of epithelioid sarcomas (ESs) show loss of integrase interactor 1 (INI1) function due to biallelic inactivation of the SMARCB1 gene on chromosome 22q11.2. Loss of INI1 is associated with overexpression of the EZH2 gene, leading to repression of tumor suppressor genes. Inhibition of EZH2, using drugs like tazemetostat, has demonstrated clinical efficacy in these patients. In fact, the FDA granted accelerated approval to tazemetostat in 2020 for treating advanced epithelioid sarcoma. Indeed, these novel small molecular and targeted inhibitors have the potential to disrupt cancer pathogenesis with limited off-target and side effects compared to traditional chemotherapy, underscoring the value of molecular profiling to not only determine treatment targets but potentially spare patients’ significant morbidity in doing so.

## 4. DNA-Based Biomarkers in STS

Due to the non-specific and overlapping histologic features of many sarcomas, molecular profiling has become essential for the diagnostic classification of soft tissue sarcomas (STSs). Beyond classification, molecular profiling also provides valuable prognostic information independent of diagnosis. As pan-cancer regulatory indications expand for immunotherapies, profiling can further determine critical metrics like tumor mutational burden (TMB) and microsatellite stability (MSI) status in sarcomas. Although whole-genome and whole-exome sequencing offer direct TMB and MSI indices, targeted assays covering key gene loci can yield similar insights. Despite the need for in-house validation, using a single molecular test for both diagnostic and therapeutic decision-making represents an efficient long-term strategy in sarcoma management.

### 4.1. Tumor Mutational Burden (TMB)

Tumor mutational burden is defined as the number of nonsynonymous coding mutations per megabase. Generally, TMB is relatively low in sarcomas compared to many carcinomas. However, some specific sarcoma subtypes, such as undifferentiated pleomorphic sarcoma (UPS), show higher-than-average TMBs compared to other STS, potentially explaining their response to immunotherapy observed in clinical trials [52,53]. Additionally, the prognostic importance of TMB may be dependent on the infiltrating immune signature in some sarcomas. RNA sequencing-derived immune profiles of 259 soft tissue sarcomas were matched with WGS-derived TMB scores. In high-TMB soft tissue sarcomas, activated T cells had a significant positive association with survival, whereas in low-TMB cases, plasma B cells were positively associated with survival [54].

It is generally believed that high TMB correlates with better outcomes following immune checkpoint inhibitor therapy, but the optimal threshold for TMB in sarcomas remains undefined as per the Association for Molecular Pathology. Because TMB is a novel metric, significant variation from laboratory-specific assays precludes a unified approach to TMB scoring and reporting. Recently, the Association for Molecular Pathology released thirteen recommendations for TMB validation studies and clinical reports [55]. In short, recommendations include transparency in reporting including the genomic regions assayed, the laboratory and bioinformatic methods used, and the quality assurance and filtering parameters applied. In summary, robust molecular profiling of STS with whole-genome, whole-exome, or large panel-targeted exome sequencing can simultaneously provide clinically useful diagnostic information as well as data for further research on the TMB of different sarcomas.

### 4.2. Microsatellite Instability (MSI)

The incidence of microsatellite instability-high (MSI-H) and mismatch repair deficiency (MMRd) in STS varies across studies, with larger studies suggesting an incidence rate of 1–2%. However, the incidence of MSI-H diagnoses in STS may depend on the methods used to profile MSI status. In one meta-analysis of 7958 cases, the authors found that MSI-H incidence in STS was 0.4% (25 of 6649 cases) by NGS, 3.1% (38 of 1222 cases) by IHC, and 7.3% (21 of 285 cases) by PCR; the meta-analysis also reported that while 194 cases were evaluated by both IHC and PCR with only four discordant results, no articles describing NGS testing reported a second method of analysis [56]. Additionally, some studies have indicated that MSI-H positively correlates with a higher TMB [57], though this is not always the case [58]. Nevertheless, cancers, including sarcomas, with either high TMB, MSI-H status, or a combination typically respond well to immunotherapy [56,59,60], underscoring their importance as predictive biomarkers. However, the role of MSI-H/MMRd as predictive biomarkers for immunotherapy in sarcomas is insufficiently explored relative to carcinomas and further study is warranted.

## 5. RNA-Based Biomarkers in STS

While DNA-based molecular profiling is well suited for detecting variants such as insertions, deletions, and single nuclear polymorphisms, RNA-based profiling is better suited for the detection of pathogenic gene rearrangements, particularly in sarcomas with a simple karyotype, as previously discussed. In addition to their diagnostic importance, such gene rearrangements can also serve as prognostic markers and even therapeutic targets. Common diagnostic examples include Ewing sarcoma with the characteristic *EWSR1-FLI1* fusion, alveolar rhabdomyosarcoma distinguished from other rhabdomyosarcomas by the *PAX3/7-FOXO1* fusion, and synovial sarcoma with its pathognomonic *SS18-SSX1/2/4* fusion. These gene fusions are diagnostic and may also represent therapeutic targets, emphasizing the importance of RNA-based molecular diagnostics in managing sarcomas. While therapeutic targeting of gene fusion products in STS is a nascent area of research, recent trials have demonstrated positive results with the use of gene fusion targeted therapy [61]. The discovery of tyrosine kinases in these fusion products is particularly promising given the success of TKI therapy in other types of genomic alterations involving tyrosine kinases [62]. In addition to its utility in detecting gene fusion products, RNA-Seq can be used to infer expression patterns of certain pathogenic proteins and immune tumor microenvironments, which in turn can be used to predict response to chemo- or immunotherapies [63].

## 6. Conclusions

The integration of genomic and molecular techniques into clinical practice has revolutionized the diagnosis, classification, and treatment of soft tissue sarcomas. Advances in DNA- and RNA-based assays, along with targeted therapies, offer personalized approaches to patient management, ultimately improving outcomes for this diverse group of malignancies. While this review aims to be comprehensive and current, genomic techniques and molecular pathology are a rapidly evolving field. Recent advancements in artificial intelligence (AI) and machine learning (ML) have the potential to revolutionize analytic methods and improve the understanding and incorporation of specific genetic alterations into greater disease pathophysiology, though specific AI/ML techniques are beyond the scope of this review. Continued research and validation of molecular biomarkers are essential to refining these strategies further and establishing new standards in sarcoma care.

## Figures and Tables

**Table 1 cancers-17-01215-t001:** Summary of molecular methods in soft tissue sarcoma and other cancers.

Technique	Genetic Material	Strengths	Pitfalls	Pivotal Studies	Study Type	Clinical Use
WGS	DNA	Widest genetic coverage	Expensive, bioinformatically intensive, covers noncoding genetic regions	Schipper et al. [4]	Prospective	Sarcoma diagnosis, theragnosis
				Watkins et al. [5]	Prospective	Sarcoma diagnosis, theragnosis
WES	DNA	Wide genetic coverage, focused on exonic genetic regions	Expensive, bioinformatically intensive, misses noncoding but not necessarily nonfunctioning (e.g., promoter/suppressor) genetic regions	Chen et al. [6]	Retrospective	Uterine leiomyosarcoma theragnosis
				Byun et al. [7]	Retrospective	Kaposi sarcoma pathogenesis
WTS	RNA	Wide transcriptomic coverage, focused on actively transcribed genetic products	Expensive, bioinformatically intensive, misses noncoding but not necessarily nonfunctioning (e.g., promoter/suppressor) genetic regions	Lorenzi et al. [8]	Retrospective	Follicular dendritic cell sarcoma diagnosis
				Astolfi et al. [9]	Retrospective	Kidney clear cell sarcoma diagnosis
				Panagopoulos et al. [10]	Retrospective	Spindle cell sarcoma pathogenesis
WGTS	DNA and RNA	Widest genetic coverage	Expensive, bioinformatically intensive, covers noncoding genetic regions	Nord et al. [11]	Retrospective	Sarcoma pathogenesis
Targeted	DNA or RNA	Targeted genetic coverage, inexpensive, bioinformatically simple	Limited only to known/targeted genes covered in particular panels	Chibon et al. [12]	Retrospective	Sarcoma prognosis
				Gounder et al. [3]	Retrospective	Sarcoma diagnosis, theragnosis

**Table 2 cancers-17-01215-t002:** Commercial liquid biopsies with US FDA approval or pending approval.

Product	Approval Status	Indications	Primary Tumor Agnostic	Panel Size	Clinical Uses
FoundationOne LiquidDx	Approved	Solid tumors	Yes	300 genes	Diagnosis and Theragnosis
Guardant 360 CDXC	Approved	Solid tumors	Yes	74 genes	Diagnosis and Theragnosis
Signatera	Pending	Bespoke/patient-specific	No	16 genes	Treatment response and Disease Monitoring
CancerSEEK	Approved	Ovary, Liver, Stomach, Pancreas, Esophagus, Colorectum, Lung, Breast	Yes	16 genes	Screening
Epi proColon	Approved	Colorectal carcinoma	Yes	1 gene	Screening

**Table 3 cancers-17-01215-t003:** Active or recently completed clinical trials using cfDNA or ctDNA in sarcoma management.

Trial	Disease	Use Case	Study Type	Phase	Brief Summary
NCT02547376	Soft tissue sarcoma	Pathogenesis	Observational	Pre-I	cfDNA as an investigatory tool to detect telomere maintenance mechanism mutations
NCT05366881	Pan-cancer	Diagnosis, disease monitoring	Observational case–control	Pre-I	Methylation of cfDNA to detect cancer early and monitor disease progression
NCT03390946	Osteosarcoma	Prognosis, Treatment response	Interventional	II	cfDNA as a predictive/prognostic biomarker for compressed chemotherapy regimen
NCT03336554	Osteosarcoma	Disease monitoring, treatment response	Observational case–control	Pre-I	Epigenetic profiling of cfDNA to determine biomarkers of disease
NCT06239272	Non-rhabdomyosarcoma soft tissue sarcoma	Diagnosis, treatment response	Interventional	I/II	Correlating mutations detected in cfDNA and tumor samples; use of cfDNA to track treatment response
NCT01772771	Pan-cancer	Diagnostic inclusion criteria	Observational	Pre-I	Correlating mutations detected in cfDNA and tumor samples
NCT03919539	Osteosarcoma	Prognosis	Observational	Pre-I	Use of cfDNA to determine mechanisms of resistance to therapy
NCT02567435	Rhabdomyosarcoma	Diagnosis	Interventional	III	Correlating mutations detected in cfDNA and tumor samples

**Table 4 cancers-17-01215-t004:** Commercial targeted RNA panels with US FDA approval or pending approval.

Product	Approval Status	Indication	Panel Size	Clinical Use
FoundationOne RNA	Approved	Solid tumors	318 genes	Diagnosis and Theragnosis
Archer FusionPlex Sarcoma v2	Research Use Only	Solid tumors	63 genes	–

**Table 5 cancers-17-01215-t005:** Spatial transcriptomic profiling methods.

Method	Resolution	Throughput	Advantages
Microdissection	Cellular–regional	Low	Manual identification of cells/regions of interest
In situ hybridization/sequencing	Subcellular–cellular	Low–medium	Highly sensitive to low RNA quantity
Spatial indexing	Subcellular–cellular	High	Capable of detecting total mRNA in tissue

## Data Availability

Not applicable.

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
