# Peer review of "Novel Molecular Methods in Soft Tissue Sarcomas: From Diagnostics to Theragnostics"

_cancers, 2025, doi:10.3390/cancers17071215_

Round 1
Reviewer 1 Report
Comments and Suggestions for Authors
The manuscript provides a comprehensive review of the molecular methodologies used in the diagnosis, classification, and theragnostic evaluation of soft tissue sarcomas (STS). It highlights the importance of genomic and transcriptomic profiling, including techniques such as whole genome sequencing (WGS), whole exome sequencing (WES), targeted sequencing, RNA sequencing (RNA-seq), and methylation analysis. The article also explores the role of circulating tumor DNA (ctDNA) and cell-free DNA (cfDNA) in liquid biopsies, as well as emerging technologies like spatial transcriptomics. Additionally, it discusses molecularly driven targeted therapies, DNA-based biomarkers (TMB, MSI), and RNA-based biomarkers that could serve as diagnostic and therapeutic targets.
The manuscript is interesting and well organzied. It could benefit from the followings:
- Consider summarizing key concepts in figures, tables, or flowcharts to enhance readability
- The review does an excellent job detailing technical methodologies but could benefit from a stronger focus on how these techniques translate into clinical practice.
- Incorporating AI and machine learning in molecular diagnostics is a growing trend—how might these enhance STS diagnosis and treatment?
- Minor grammatical errors and awkward phrasing should be revised for clarity and conciseness.
- the followings manuscript should be referenced for proper discussion:
Unveiling the Genomic Basis of Chemosensitivity in Sarcomas of the Extremities: An Integrated Approach for an Unmet Clinical Need. Int J Mol Sci. 2023 Apr 8;24(8):6926. doi: 10.3390/ijms24086926. PMID: 37108089; PMCID: PMC10138892.
6. latest WHo should be included
7 study limitations should be also summarized
Author Response
We thank the reviewers for their through review. Below is a point by point to the reviewer’s comments
Reviewer 1
- Consider summarizing key concepts in figures, tables, or flowcharts to enhance readability
Response: major conclusions and summaries of each section are included in the five tables already included in the manuscript. While this review discusses different genomic techniques, specific laboratory methods/Standard Operating Procedures, which are more suited to graphic representation (in figures or flowcharts), are beyond the scope of this manuscript.
- The review does an excellent job detailing technical methodologies but could benefit from a stronger focus on how these techniques translate into clinical practice.
Response: translation of techniques into clinical practice is references throughout the manuscript, for example: discussion of ct/dfDNA as a minimally invasive biopsy for initial diagnosis or disease monitoring in currently ongoing clinical trials (section 2.1.1); machine learning of methylation analysis to generate a sarcoma classifier as an assistive diagnostic tool for challenging cases (section 2.3); the use of small molecule inhibitor therapy for specific fusion positive sarcomas (section 3); quantification of tumor mutational burden and microsatellite instability by molecular profiling (sections 4.1, 4.2).
- Incorporating AI and machine learning in molecular diagnostics is a growing trend—how might these enhance STS diagnosis and treatment?
Response: AI and machine learning are emerging technologies with significant promise to revolutionize analytic methods and understanding of new discoveries to disease processes. Some discussion around machine learning as a tool for parsing genomic data is given already, such as in methylation analysis (section 2.3). However, specific AI and machine learning techniques are beyond the scope of this text, which focuses primarily on sequencing technologies and their clinical uses.
- Minor grammatical errors and awkward phrasing should be revised for clarity and conciseness.
Response: Manuscript grammar and phrasing has been revised for clarity and conciseness.
- the followings manuscript should be referenced for proper discussion:
Unveiling the Genomic Basis of Chemosensitivity in Sarcomas of the Extremities: An Integrated Approach for an Unmet Clinical Need. Int J Mol Sci. 2023 Apr 8;24(8):6926. doi: 10.3390/ijms24086926. PMID: 37108089; PMCID: PMC10138892.
Response: the reference has been included in the text.
- latest WHo should be included
Response: Reference to the latest WHO and molecularly-defined soft tissue sarcomas is made in the manuscript (section 2).
- study limitations should be also summarized
Response: Study limitations are added to the conclusion (section 6).
Reviewer 2 Report
Comments and Suggestions for Authors
Thank you for the opportunity to review the paper titled “Novel molecular methods in soft tissue sarcomas: from diagnostics to theragnostics”. In this review, the authors present updates on the most recent molecular techniques employed in sarcoma diagnosis and theragnosis, providing a broad and in-depth discussion based on technique types. While the paper generally meets the journal’s requirements, I have several suggestions for improving its clarity, coherence and overall quality.
- Abbreviations should be defined when first introduced. Please add WTS after whole transcriptome sequencing on Page 2, Line 61.
- Authors mentioned that “each technique has its strengths and pitfalls as summarized in Table 1” on Page 2 Line 63, however, the table does not include any discussion of the strengths and pitfalls.
- There is some confusion in Section 2.1.1 regarding ctDNA/cfDNA. In this part, ctDNA is defined as tumor genetic fragments released into the bloodstream, however, cfDNA is never explicitly defined. Additionally, all the clinical trials listed in Table 3 appear to be cfDNA-based. Are there any clinical trials specially focused on ctDNA? Furthermore, are ctDNA and cfDNA interchangeable in the clinical setting?
- I understand that authors structure this review based on the framework of “General techniques-DNA markers in STS-RNA markers in STS”. However, in my view, the article’s structure could be more cohesive, as the discussion of molecular techniques includes considerable details on their application in STS in “General techniques” part (Section 2. Current Molecular Techniques in Sarcoma Diagnosis and Theragnosis). For instance, the authors discuss RNA-seq as a practical method for detecting fusion genes in sarcoma and provide several examples to support this point in Section 2, then why is RNA-based profiling discussed again in Section 5? Similarly, CSF1 is a fusion component of CSF1-COL6A3 in tenosynovial giant cell tumor, and why is this topic on fusion genes discussed in a separate section (Section 3. Molecularly Driven Targeted Therapies in STS) rather than being integrated with related discussions? I suggest consolidating related argruments instead of presenting them across similar sections. The same applies to Section 4 on DNA-based biomarkers. Have the authors considered integrating this discussion into the section on DNA techniques?
Author Response
- Abbreviations should be defined when first introduced. Please add WTS after whole transcriptome sequencing on Page 2, Line 61.
Response: Abbreviation added.
- Authors mentioned that “each technique has its strengths and pitfalls as summarized in Table 1” on Page 2 Line 63, however, the table does not include any discussion of the strengths and pitfalls.
Reponse: Table 1 has been expanded to include summarized strengths and pitfalls of each technique
- There is some confusion in Section 2.1.1 regarding ctDNA/cfDNA. In this part, ctDNA is defined as tumor genetic fragments released into the bloodstream, however, cfDNA is never explicitly defined. Additionally, all the clinical trials listed in Table 3 appear to be cfDNA-based. Are there any clinical trials specially focused on ctDNA? Furthermore, are ctDNA and cfDNA interchangeable in the clinical setting?
Response: We have clarified in the manuscript in section 2.1.1 the definitions of cfDNA (circulating DNA recovered from any cell, benign or malignant) and ctDNA(circulating DNA from only malignant cells). We have provided some non-cancer examples of the use of cfDNA for definitional purposes. We have also clarified that for the purpose of this review, which is entirely malignancy focused, the two terms are interchangeable.
- I understand that authors structure this review based on the framework of “General techniques-DNA markers in STS-RNA markers in STS”. However, in my view, the article’s structure could be more cohesive, as the discussion of molecular techniques includes considerable details on their application in STS in “General techniques” part (Section 2. Current Molecular Techniques in Sarcoma Diagnosis and Theragnosis). For instance, the authors discuss RNA-seq as a practical method for detecting fusion genes in sarcoma and provide several examples to support this point in Section 2, then why is RNA-based profiling discussed again in Section 5? Similarly, CSF1 is a fusion component of CSF1-COL6A3 in tenosynovial giant cell tumor, and why is this topic on fusion genes discussed in a separate section (Section 3. Molecularly Driven Targeted Therapies in STS) rather than being integrated with related discussions? I suggest consolidating related argruments instead of presenting them across similar sections. The same applies to Section 4 on DNA-based biomarkers. Have the authors considered integrating this discussion into the section on DNA techniques?
Round 2
Reviewer 1 Report
Comments and Suggestions for Authors
Ref 63 is correcly reporter in the text but not in the references
minor revisions
Author Response
Edits are done as per reviewer's comments.